# Engineering Process Optimization and Quality Stability Control of High-Speed Laser Cladding Coatings Based on AHP-FCE

Yifei Xv [1], Yaoning Sun [1,*], Wangjun Cheng [1] and Yuhang Zhang [1,2,*]

1 School of Mechanical Engineering, Xinjiang University, Urumqi 830017, China; xyf124@stu.xju.edu.cn (Y.X.); chengwangjun2008@126.com (W.C.)

2 Institute of New Materials, Guangdong Academy of Sciences, National Engineering Laboratory for Modern Materials Surface Engineering Technology, Guangdong Provincial Key Laboratory of Modern Surface Engineering Technology, Guangzhou 510651, China

* Correspondence: synzxd2000@xju.edu.cn (Y.S.); zyh1996123@stu.xju.edu.cn (Y.Z.)

**Abstract:** Due to the rapid advancement in processing efficiency, high-speed laser cladding has demonstrated significant potential in the repair and protection of various substrates. In this study, we established a comprehensive evaluation model for the coating quality of Fe-Cr-Ni-based alloy with high-speed laser cladding using the analytic hierarchy process and fuzzy comprehensive evaluation method (AHP-FCE). The weights obtained through the analytic hierarchy process for forming quality, microstructure, and surface performance are as follows: $W_{B1} = 0.1365$, $W_{B2} = 0.2385$, and $W_{B3} = 0.625$, respectively. During the fuzzy comprehensive evaluation step, an evaluation level was graded while quantifying the level range through membership function judgment. By combining subjective and objective evaluations, qualitative issues were transformed into quantitative assessment methods. Through comprehensive evaluation analysis, it was concluded that the scanning speed of high-speed laser cladding had a greater impact on coating thickness compared to powder feeding speed while significantly enhancing microstructure densification. The overlap rate exerted the most influence on dilution rate homogenization of near-surface dendrites. Simultaneously, the optimal preparation technology was determined: laser power 660 W, scanning speed 14,400 mm/min, overlap rate/min. This study transforms multi-objective quality evaluation of high-speed laser cladding coatings into a single objective problem by realizing comprehensive quality quantification and providing a new method for quantitative evaluation and visualization of coating quality.

**Keywords:** high-speed laser cladding; process optimization; AHP-FCE; coating quality stability control

## 1. Introduction

High-speed laser cladding is a next-generation processing technology with a fast processing speed, high processing accuracy, and low thermal impact on the substrate [1,2]. It is widely used to repair and protect oil drill parts to overcome the low efficiency of traditional laser cladding repair [3,4]. With the rapid increase of laser beam scanning speed, the high overlap between paths and the two-stage melting process before and after powder material enters the molten pool are common in high-speed laser cladding coatings [5–7]. As a result, the coating quality, including its forming quality, near-surface microstructure, and surface properties, is significantly different from those produced using traditional laser cladding [8–10].

Different application environments have different requirements and evaluation criteria for the quality of coatings. Existing studies on the quality of high-speed laser cladding coatings have mainly focused on quantitative and qualitative analyses of single-pass cladding molding (layer cladding width, layer cladding height, width-to-height ratio,

dilution rate, and wetting angle) and have rarely investigated multi-path laser cladding coatings. In order to investigate the effects of laser power, scanning speed, and powder feeding speed on the track geometry of YCF104 clad tracks, Zhao [11] designed 125 sets of single-factor experiments to examine their impact on the cross-sectional area, width, and depth of the track as well as the width and depth of the heat-affected zone. Menghani [12] employed a full factorial design approach to briefly explore the overlap rate, microhardness, and microstructure of individual coatings while determining optimal cladding conditions through multi-response optimization. Khorram [13,14], respectively utilizing response surface methodology and central composite design for cladding 718 CrC + 75 (25Ni80) coating, investigated its influence on geometric parameters (width, height, and cladding angle), dilution rate, and hardness. These output responses effectively predict the cladding process. Chen [15] conducted multi-pass lap experiments using the Taguchi method to obtain geometric properties (coating thickness, coating width) and mechanical properties (microhardness) of coatings. Based on experimental results analyzed by signal-to-noise ratio (S/N), important factors influencing quality characteristics were identified with subsequent selection of optimal process parameters. These studies also did not comprehensively evaluate the factors of coatings, and there are unclear boundaries of the relationship between evaluation results and process parameters.

The analytical hierarchy process (AHP) method is a multi-objective decision analysis methodology proposed by T.L. Satty [16–18]. It can organize problems into a physical and hierarchical structure to determine the weights of different factors by constructing a hierarchical structure model for analysis [19]. Fuzzy comprehensive evaluation (FCE) was proposed by the American cyberneticist L.A. Zadeh in 1965 [20,21]. It is a method to quantitatively evaluate qualitative problems [22,23]. Analytic hierarchy process-fuzzy comprehensive evaluation (AHP-FCE) [24,25] combines AHP with FCE and has been widely used for analysis of the corrosion failure mechanism of equipment [26,27], life prediction and risk assessment [28,29], risk analysis of mine engineering construction [30–32], and evaluation of equipment design scheme [33,34]. Wang [35] employed response surface methodology based on a composite center design in studying processing parameter effects on morphology and quality in Fe1045/TiC multipass laser cladding onto AISI medium carbon steel. Multiple responses such as composite width, flatness, and non-fusion area were transformed into a single target correlation analysis while weights were objectively determined through principal component analysis; however, the comprehensive quantification of coating quality has not been achieved.

In this study, 25 sets of experiments were designed, using the Taguchi method to prepare coatings by high-speed laser cladding technology. The layer cladding height (H), molten pool depth (D), dilution rate ($\eta$), dendrite size (Ds), surface roughness (Ra), and microhardness (HV0.2) of the coating were characterized. Through fixed evaluation methods and quantitative indexes, the stable microstructure and properties of the coating can be ensured within a certain process range. Based on these evaluation indexes, a comprehensive evaluation model of coating quality was established by the AHP-FCE method, and the commonly used multi-objective evaluation process was transformed into a single problem. The establishment of comprehensive evaluation model realizes the quantification and visualization of coating quality evaluation. At the same time, it provides a reference for the quality evaluation of a high-speed laser cladding layer and the selection of process parameters.

## 2. Experiment

### 2.1. Experimental Parameters and Materials

An ASTM 1045 steel plate with dimensions of 70 mm × 150 mm × 8 mm was used as the substrate. Before the cladding experiment, the surface of the substrate was ground with 400 grit abrasive papers, washed with acetone, and dried for use. The surface roughness of the substrate after grinding was determined to be 7.0–8.0 μm. In this experiment, to fabricate coating on ASTM 1045 steel substrate, Fe-Cr-Ni-based alloy powders (35–53 μm,

Nanjing Zhongke Yuchen Co., Ltd., Nanjing, China) were selected as cladding materials. The chemical composition of the powders is listed in Table 1. The macroscopic morphology of the sample is shown in Figure 1.

**Table 1.** Chemical composition of the Fe-Cr-Ni-based alloy powders (wt. %).

| C | Si | Cr | Ni | Mo | B | Fe |
|---|---|---|---|---|---|---|
| 0.15 | 4.5 | 22 | 13 | 2 | 1.6 | Bal. |

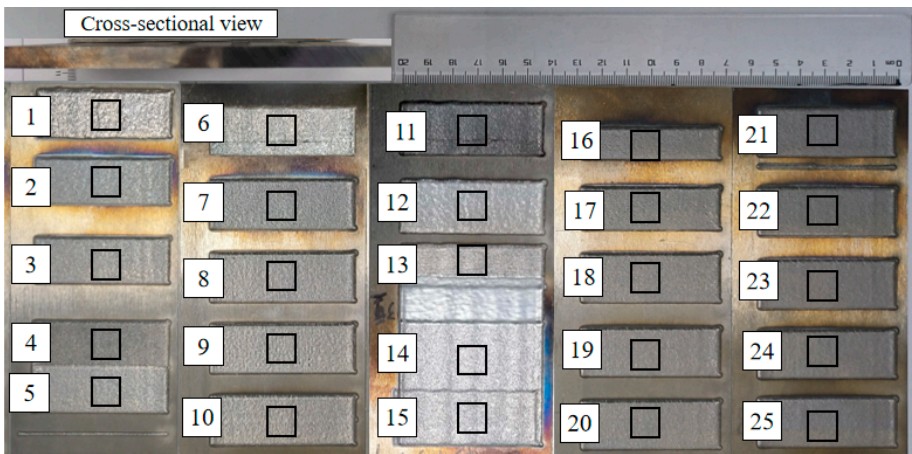

**Figure 1.** Macroscopic morphology of high-speed laser cladding coating.

The cladding experiment was carried out on a ZKZM-2000 fiber high-speed laser cladding system. The experimental setup is shown in Figure 2, and its upper and lower bounds are shown in Table 2. The experimental spot diameter was 1.2 mm, with a defocus of 15 mm. Coatings were prepared on the surface of ASTM 1045 steel by coaxial powder feeding. The shielding gas and powder feeding gas were both argon (purity: 99.99%). The factors and levels of cladding experiment design are shown in Table 3. After cladding, a sample with dimensions of 10 mm × 10 mm × 8 mm was cut perpendicular to the scanning direction. After grinding and polishing, the metallographic samples were obtained by etching in aqua regia ($V_{HCl} : V_{HNO_3} = 3:1$).

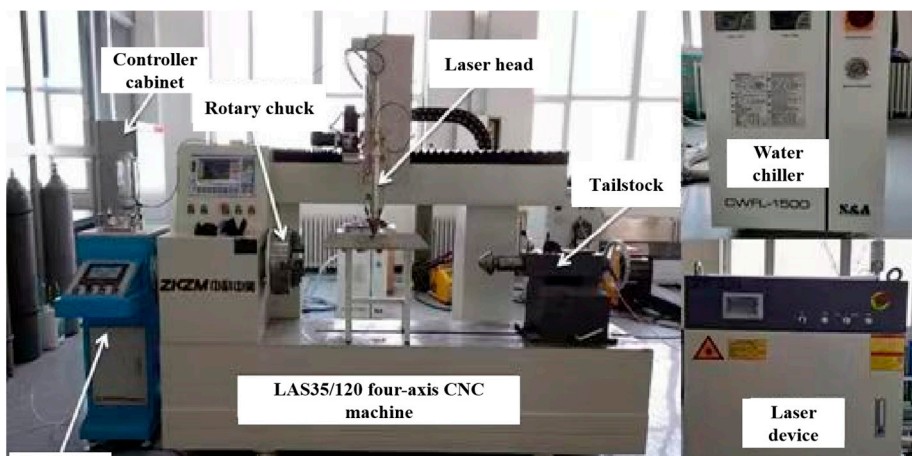

**Figure 2.** ZKZM-2000 fiber high-speed laser cladding system.

**Table 2.** The main parameters of the ZKZM-2000 fiber high-speed laser cladding system.

| Parameter Type | Inversion |
|---|---|
| Power | 500 W–2000 W |
| Wave length | 1080 nm |
| Spot diameter | 1.2 mm |
| Powder feeding method | Three-way coaxial powder feeding |
| Scanning speed | 0–20 m |
| Gas flow | 20–25 L/min |
| Maximum spindle speed | 200 r/min |
| Machine stroke (X axis) | 3222 mm |
| Machine stroke (Y-axis) | 400 mm |
| Machine stroke (Z-axis) | 300 mm |

**Table 3.** Taguchi test factors and levels.

| No. | Laser Power/ P (W) | Scanning Speed/ Ss (mm/min) | Overlap Ratio/ Or (%) | Powder Flow Rate/ Vp (r/min) |
|---|---|---|---|---|
| 1 | 660 | 3600 | 20 | 2.5 |
| 2 | 880 | 7200 | 35 | 3 |
| 3 | 1100 | 10,800 | 50 | 3.5 |
| 4 | 1320 | 14,400 | 65 | 4 |
| 5 | 1540 | 18,000 | 80 | 4.5 |

In traditional laser cladding, a high-energy laser beam simultaneously melts the substrate material and powder particles to form a melt pool. Because the matrix absorbs more energy, the temperature of the melt pool is higher than that of the powder particles, Tp. In contrast, high-speed laser cladding heats powder particles close to their melting point and then sprays the substrate surface at a high speed to form an extremely thin metallurgical layer after a short contact with the melt pool on the substrate. The temperature of the melt pool is almost the same as that of the powder particles ($T_{liq} \approx T_p$) [36]. Because the substrate absorbs less energy, the heat-affected zone is smaller. The working principles of both traditional laser cladding and high-speed laser cladding are shown in Figure 3.

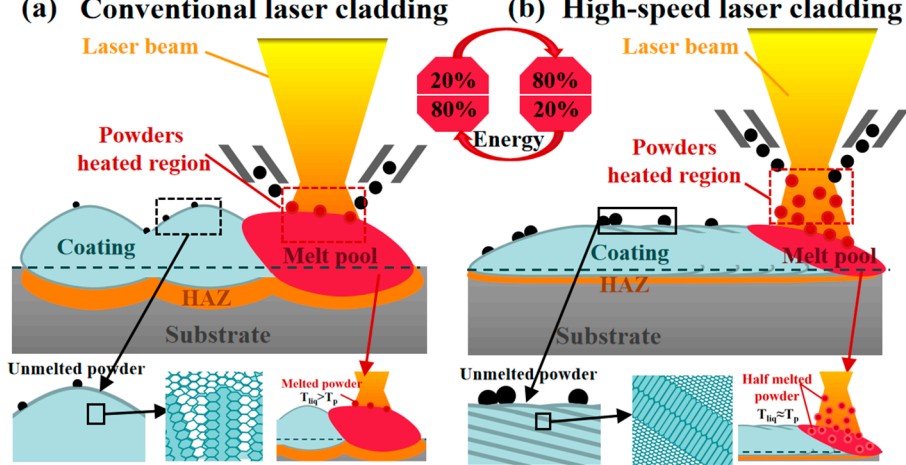

**Figure 3.** Working principle diagram; (**a**) Conventional laser cladding; (**b**) High-speed laser cladding.

## 2.2. Experimental Characterization

The cross-section of the cladding layer was observed using light microscopy to measure the height of the cladding layer and the depth of the melt pool (measurements were taken at three positions for each sample). The surface roughness was measured with a hyper-depth microscopy system. The microhardness of the coating was measured by a Vickers

microhardness tester (load: 200 g, loading time: 15 s, measurement position: 50 μm from the surface at the same height), and the average value at three test points was taken. The microstructure of the coating was observed using scanning electron microscopy (SEM). The dendrite size statistics were analyzed using ImageJ software (1.8.0.345) on an SEM map of the near-surface position of the coating at a 5000-times magnification. Figure 4 shows the measurement method of each index.

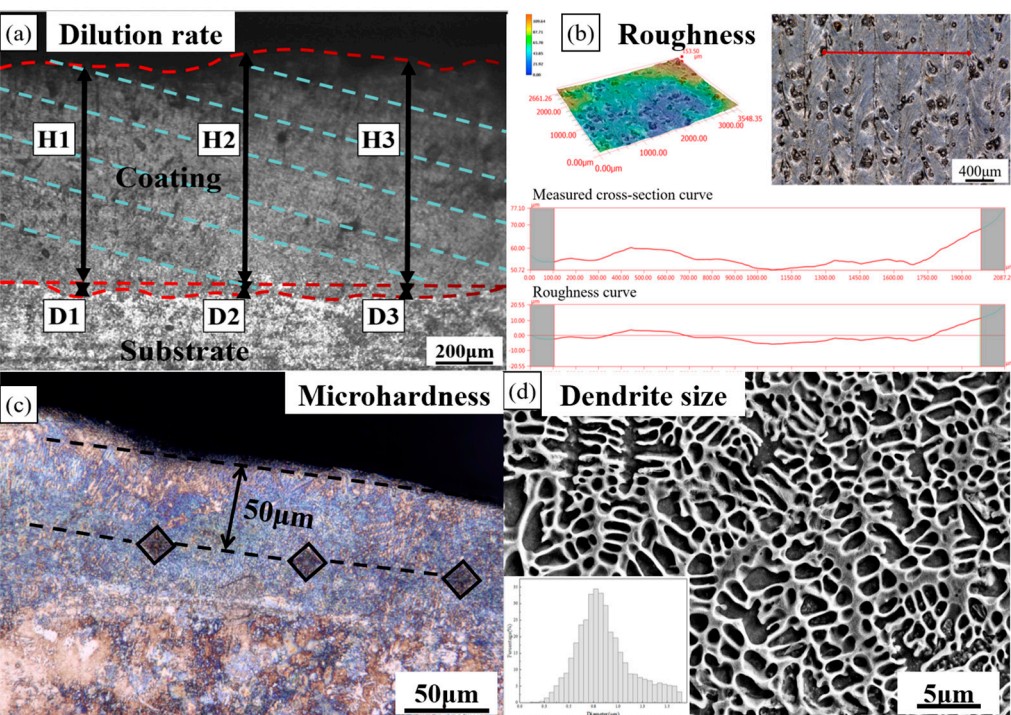

**Figure 4.** Measurement methods of each index; (**a**) Layer cladding height (H) and molten pool depth (D); (**b**) Surface roughness; (**c**) Microhardness; (**d**) Dendrite size.

### 2.3. Experimental Characterization

A high overlap rate is usually necessary to obtain sufficiently dense, thick coatings using high-speed laser cladding. Because each cladding layer is thin and wide, there are far fewer coating surface fluctuations than in traditional laser cladding. Combining Figures 3 and 4a shows that the high overlap rate of high-speed laser cladding produced minimal fluctuations on the bonding surface between the coating and substrate. The dilution rate η was calculated as Equation (1):

$$\eta \approx \frac{\sum_{i=1}^{n} \frac{D_i}{D_i + H_i}}{n}, \tag{1}$$

where $D_i$ and $H_i$ are the melt depth and coating height on the *i*-th path, respectively, and *n* is the total number of melting ways.

Statistical analysis of the layer cladding height, molten pool depth, and dilution rate of 25 samples (Figure 5) showed that layer cladding height changed greatly depending on the scanning speed. The molten pool depth changed only slightly when using different processing parameters. Table 4 shows the micro-morphology of the sample surface at 200-times magnification, which shows that the coating surface roughness (Ra) was affected by both the scanning speed (Ss) and overlap rate (Or). An overlap rate of 50% was a critical value, below which the coating fluctuation is obvious. When the overlap rate was greater than 50%, the overlap trace gradually disappeared, and the main influencing factor of surface roughness changed from the size of lap fluctuations to the degree of powder melting.

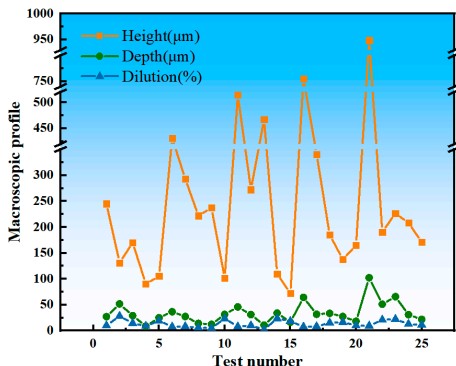

**Figure 5.** Layer cladding height, molten pool depth, and dilution rate of high-speed laser cladding coating.

**Table 4.** Surface microscopic morphology and roughness of high-speed laser cladding coating.

| P (W) \ Ss (mm/min) | 3600 | 7200 | 10,800 | 14,400 | 18,000 |
|---|---|---|---|---|---|
| 660 | Ra=34.63 [1] <br> Or = 20% | Ra=25.63 [2] <br> Or = 35% | Ra=6.71 [3] <br> Or = 50% | Ra=3.03 [4] <br> Or = 65% | Ra=9.96 [5] <br> Or = 80% |
| 880 | Ra=26.63 [6] <br> Or = 35% | Ra=24.8 [7] <br> Or = 50% | Ra=4.49 [8] <br> Or = 65% | Ra=4.62 [9] <br> Or = 80% | Ra=24.05 [10] <br> Or = 20% |
| 1100 | Ra=25.75 [11] <br> Or = 50% | Ra=17.9 [12] <br> Or = 65% | Ra=10.75 [13] <br> Or = 80% | Ra=22.2 [14] <br> Or = 20% | Ra=14.26 [15] <br> Or = 35% |
| 1320 | Ra=18.9 [16] <br> Or = 65% | Ra=18.55 [17] <br> Or = 80% | Ra=21.38 [18] <br> Or = 20% | Ra=20.89 [19] <br> Or = 35% | Ra=18.82 [20] <br> Or = 50% |
| 1540 | Ra=20.57 [21] <br> Or = 80% | Ra=26.57 [22] <br> Or = 20% | Ra=17.65 [23] <br> Or = 35% | Ra=4.49 [24] <br> Or = 50% | Ra=5.53 [25] <br> Or = 65% |

High-speed laser cladding coatings typically show a multi-pass lap structure with a small melt pool, which produces a uniform micro-structure. They include three main areas: the inlet surface area, the middle multi-layer lap area, and the bottom/matrix interface bonding area. The size of dendrites on the melted layer relates to the growth rate of dendrites, cooling rate, temperature gradient, and local solidification time [37–39]. Figure 6 shows a correlation between the microhardness and dendrite size in the near-surface region of the sample, in which the finer the grain and the denser the structure, the greater the hardness of the coating. Table 5 shows the micro-structure diagram of the near-surface

position of a selected section of the sample's coating. The micro-structure was mainly affected by the overlap rate, and repeated heat treatment caused by the high overlap rate may have homogenized the grains.

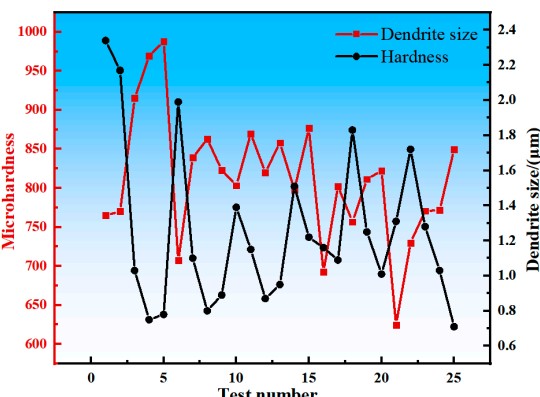

**Figure 6.** Microhardness and dendrite size of high-speed laser cladding coating near surface position.

**Table 5.** Micro-structure of high-speed laser cladding coating section near surface position.

| P (W) \ Ss (mm/min) | 3600 | 7200 | 10,800 | 14,400 | 18,000 |
|---|---|---|---|---|---|
| 660 | 1<br>Or = 20% | 2<br>Or = 35% | 3<br>Or = 50% | 4<br>Or = 65% | 5<br>Or = 80% |
| 880 | 6<br>Or = 35% | 7<br>Or = 50% | 8<br>Or = 65% | 9<br>Or = 80% | 10<br>Or = 20% |
| 1100 | 11<br>Or = 50% | 12<br>Or = 65% | 13<br>Or = 80% | 14<br>Or = 20% | 15<br>Or = 35% |
| 1320 | 16<br>Or = 65% | 17<br>Or = 80% | 18<br>Or = 20% | 19<br>Or = 35% | 20<br>Or = 50% |
| 1540 | 21<br>Or = 80% | 22<br>Or = 20% | 23<br>Or = 35% | 24<br>Or = 50% | 25<br>Or = 60% |

## 3. Evaluation Methodology

The AHP-FCE method was used to evaluate the quality of high-speed laser cladding coatings using the process shown in Figure 7. First, the quality index of high-speed laser cladding coating was determined, and the corresponding layered structure was established. Second, the weights of the final indicators were calculated based on the AHP. Finally, these

indicators were introduced into the fuzzy comprehensive evaluation model to determine the final quality evaluation level. This section details the overall AHP-FCE evaluation process.

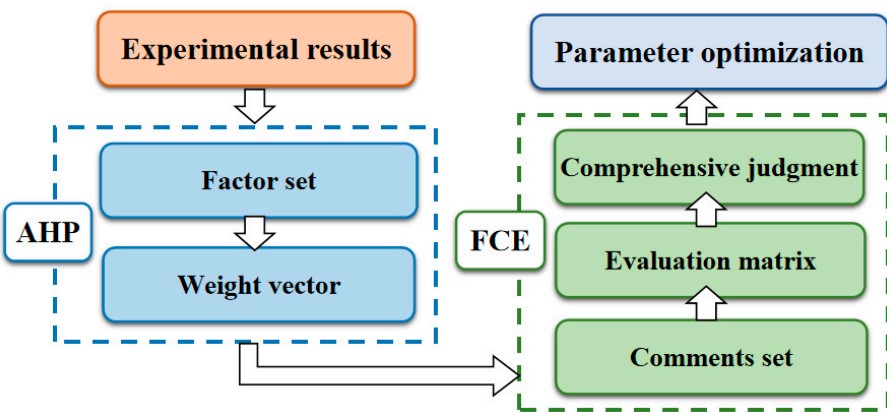

**Figure 7.** Quality evaluation process of high-speed laser cladding coating by the AHP-FCE method.

*3.1. Evaluation Indicators for the Quality of High-Speed Laser Cladding Coating*

We evaluated the quality of the high-speed laser cladding coating in terms of the forming quality, microstructure, and surface properties. The coating forming quality included layer cladding height (H), molten pool depth (D), and the dilution rate (η). The near-surface microstructure was characterized in terms of the dendrite size (Ds). The surface properties included the surface roughness (Ra) and microhardness (HV0.2). According to the relationship between the above quality indicators, a hierarchical structure was established, as shown in Figure 8.

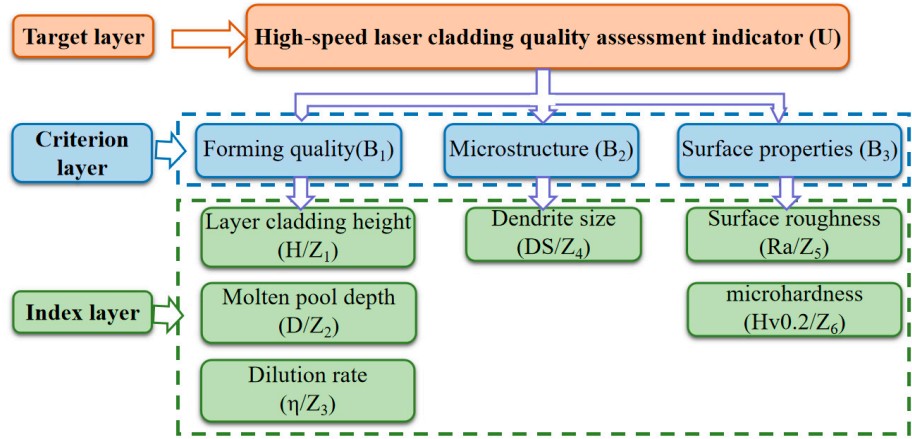

**Figure 8.** Hierarchy of quality indicators for high-speed laser cladding coatings.

*3.2. Weight Vectors Determined by AHP*

The problem to be solved by the analytic hierarchy process (AHP) is the weight of the lower layer relative to the higher layer. AHP provides a qualitative and quantitative analysis method to solve complex multi-objective problems [40]. The steps are as follows:

3.2.1. Build the Hierarchy

As shown in Figure 8, a logical hierarchy of evaluation indexes was constructed for the quality of high-speed laser cladding layers. It consisted of a target layer (U), a criterion layer (classification $B_i$), and an indicator layer (element $Z_j$).

3.2.2. Establish the Judgment Matrix

The Saaty scale in Table 6 was used to construct the judgment matrix. We compared pairs of elements at the same level.

**Table 6.** Saaty scale.

| Digital Scale | Inversion | Definition |
|:---:|:---:|:---:|
| 1 | 1 | Equally important |
| 3 | 1/3 | Moderately important |
| 5 | 1/5 | Strongly important |
| 7 | 1/7 | Extremely important |
| 9 | 1/9 | Completely important |
| 2, 4, 6, 8 | 1/2, 1/4, 1/6, 1/8 | Intermediate value of the above importance |

The judgment matrix is shown in Equation (2). Referring to Figure 7 shows that four judgment matrices need to be established in this work.

$$A = \begin{bmatrix} a_{11} & a_{12} & \cdots & a_{1k} \\ a_{21} & a_{22} & \cdots & a_{2k} \\ \vdots & \vdots & a_{ij} & \vdots \\ a_{k1} & a_{k2} & \cdots & a_{kk} \end{bmatrix}, (i = 1, 2, \cdots, k, j = 1, 2, \cdots, k), \tag{2}$$

where $a_{ij}$ is the importance degree of factor i to factor j, which should be satisfied as described in Equation (3).

$$a_{ij} = \frac{1}{a_{ji}}, a_{ij} = 1 (i = j), \tag{3}$$

$$U = \{B_1, B_2, B_3\} = \begin{bmatrix} 1 & \frac{1}{2} & \frac{1}{4} \\ 2 & 1 & \frac{1}{3} \\ 4 & 3 & 1 \end{bmatrix}; B_1 = \begin{bmatrix} 1 & 1 & \frac{1}{3} \\ 1 & 1 & \frac{1}{3} \\ 3 & 3 & 1 \end{bmatrix}, B_2 = [1], B_3 = \begin{bmatrix} 1 & \frac{1}{2} \\ 2 & 1 \end{bmatrix}$$

### 3.2.3. Consistency Check

The judgment matrix is highly subjective and requires consistency checking to ensure that its error is within an acceptable range. The results of the consistency test require the CR value (CR = CI/RI) to be less than 0.1. Table 7 shows the random index (RI).

$$CI = \frac{\lambda_{max} - n}{n - 1}, \tag{4}$$

where $\lambda_{max}$ is the largest eigenvalue of the judgment matrix, and *n* is the number of columns or rows in the judgment matrix.

$$\lambda = 3.018, CI = \frac{\lambda - 3}{3 - 1} = 0.009, CR = \frac{CI}{RI} = \frac{0.009}{0.58} = 0.017 < 0.1$$

$$\lambda_1 = 3, CI = \frac{\lambda_{max} - 3}{3 - 1} = 0, CR = \frac{CI}{RI} = \frac{0}{0.58} = 0 < 0.1$$

**Table 7.** Random index values for different scales.

| n | 1 | 2 | 3 | 4 | 5 | 6 | 7 | 8 | 9 |
|:---:|:---:|:---:|:---:|:---:|:---:|:---:|:---:|:---:|:---:|
| RI | 0 | 0 | 0.58 | 0.89 | 1.12 | 1.26 | 1.36 | 1.41 | 1.46 |

According to the relevant theorems, the equations of order 1 and 2 are perfectly consistent. Therefore, the four judgment matrices created in this study passed the consistency test, thus ensuring the validity of all indicator weights.

### 3.2.4. Weight Calculation

After consistency detection, the weight vector was calculated by the square root method, and the results are shown in Table 8 and Figure 9.

**Table 8.** Weighted results of first-level and second-level evaluation indicators.

| General Objectives | First-Level Evaluation Indicators | $W_{Bi}$ | Second-Level Evaluation Indicators | $W_{Zj}$ | $W_i$ $(j)$ |
|---|---|---|---|---|---|
| U | $B_1$ | 0.1365 | $Z_1$ | 0.2 | 0.0273 |
| | | | $Z_2$ | 0.2 | 0.0273 |
| | | | $Z_3$ | 0.6 | 0.0819 |
| | $B_2$ | 0.2385 | $Z_4$ | 1 | 0.2385 |
| s | $B_3$ | 0.625 | $Z_5$ | 0.333 | 0.2081 |
| | | | $Z_6$ | 0.667 | 0.4169 |

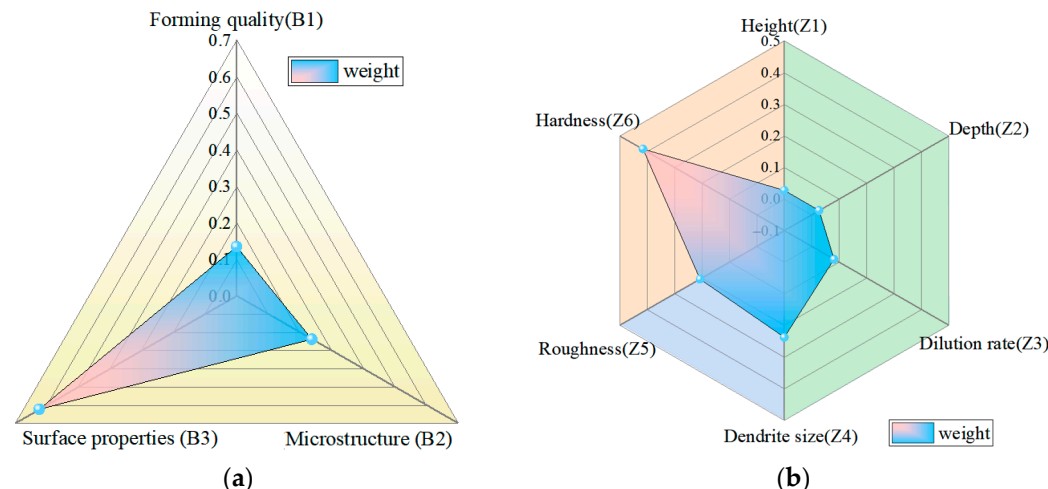

**Figure 9.** The weight of each factor in (**a**) the criterion layer and (**b**) the index layer.

### 3.3. Establishment of the FCE Model

3.3.1. Establishment of the Evaluation Indicator Set U

The classification of the set of evaluation metrics is shown in Figure 9, which includes two levels of indicators: the criterion level U = {$B_1$, $B_2$, $B_3$} and the indicator level B = {$Z_1$, $Z_2$, $Z_3$, $Z_4$, $Z_5$, $Z_6$}.

3.3.2. Establishment of the Weighting Coefficients Set W

Weights of the criterion layer W($B_1$, $B_2$, $B_3$) = (0.1365, 0.2385, 0.625).
Weights of the indicator layer W($Z_1$, $Z_2$, $Z_3$) = (0.2, 0.2, 0.6), W($Z_4$) = (1), W($Z_5$, $Z_6$) = (0.333, 0.667).

3.3.3. Establishment of Quality Evaluation Level Set V

The quality of high-speed laser cladding coatings was evaluated at four levels, also known as V = (V1, V2, V3, V4) = (excellent, good, fair, poor). The corresponding score is V = (4, 3, 2, 1). The range of grades was quantified by combining experience and reviewing the relevant literature, and the results are shown in Table 9.

**Table 9.** The value range of second-level evaluation indicators of different quality.

| Criterion Layer Evaluation Indicators | Index Layer Evaluation Indicators | V1 | V2 | V3 | V4 |
|---|---|---|---|---|---|
| Forming quality ($B_1$) | Height (μm) | 225–275 | 175–225, 275–350 | 125–175, 350–450 | ≤125, >450 |
| | Depth (μm) | 15–25 | ≤15, 25–35 | 35–45 | >45 |
| | Dilution rate (%) | 7.5–12.5 | ≤7.5, 12.5–17.5 | 17.5–22.5 | >22.5 |
| Microstructure ($B_2$) | Dendrite size (μm) | ≤0.95 | 0.95–1.25 | 1.25–1.55 | >1.55 |
| Surface properties ($B_3$) | Roughness (μm) | ≤12 | 12–20 | 20–28 | >28 |
| | Hardness | ≥870 | 810–870 | 750–810 | <750 |

### 3.3.4. Establishment of Stepwise Affiliation Function

To increase the accuracy of the membership relationship between the quality level and the measured data, the triangular stepwise membership function shown in Figure 10 was established. The quality level included two types of semi-closed intervals and closed intervals, where $\lambda_1$, $\lambda_2$, $\lambda_3$, and $\lambda_4$ were the endpoints of the interval, and $\lambda_2$ and $\lambda_3$ are also the midpoints of the closed interval. Tables 10–12 compare the membership functions of each factor in the index layer established according to Figure 10.

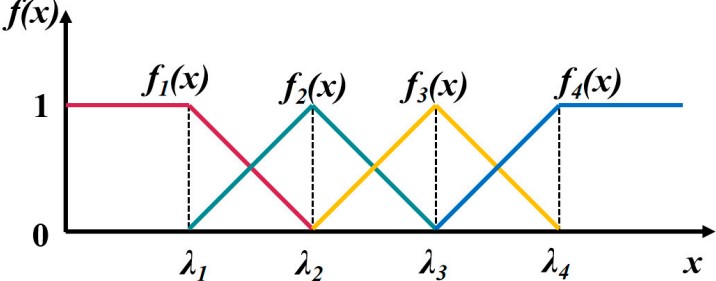

**Figure 10.** Triangular stepwise affiliation function.

**Table 10.** Judgment of height of each sample.

| Fuzzy Performance Evaluation | Membership Function | Melting Height Range | Membership Function | Melting Height Range |
|---|---|---|---|---|
| V1 | 0 | H ≤ 200 | | |
| | $(x-200)/50$ | 200 < H ≤ 250 | | |
| | $1-(x-250)/50$ | 250 < H ≤ 300 | | |
| | 0 | H > 300 | | |
| V2 | 0 | H ≤ 150 | 0 | H ≤ 250 |
| | $(x-150)/50$ | 150 < H ≤ 200 | $(x-250)/50$ | 250 < H ≤ 300 |
| | $1-(x-200)/50$ | 200 < H ≤ 250 | $1-(x-300)/100$ | 300 < H ≤ 400 |
| | 0 | H > 250 | 0 | H > 400 |
| V3 | 0 | H ≤ 100 | 0 | H ≤ 300 |
| | $(x-100)/50$ | 100 < H ≤ 150 | $(x-300)/100$ | 300 < H ≤ 400 |
| | $1-(x-150)/50$ | 150 < H ≤ 200 | $1-(x-400)/100$ | 400 < H ≤ 500 |
| | 0 | H > 200 | 0 | H > 500 |
| V4 | 1 | H ≤ 100 | 0 | H ≤ 400 |
| | $1-(x-100)/50$ | 100 < H ≤ 150 | $(x-400)/100$ | 400 < H ≤ 500 |
| | 0 | H > 150 | 1 | H > 500 |

**Table 11.** Judgment of depth, dilution rate, and dendrite size of each sample.

| Fuzzy Performance Evaluation | Membership Function | Penetration Range | Membership Function | Dilution Rate Range | Membership Function | Grain Scale Inch Range |
|---|---|---|---|---|---|---|
| V1 | 0<br>$(x-10)/10$<br>$1-(x-20)/10$<br>0 | $0 < D \leq 10$<br>$10 < D \leq 20$<br>$20 < D \leq 30$<br>$D > 30$ | 0<br>$(x-5)/5$<br>$1-(x-10)/5$<br>0 | $0 < \eta \leq 5$<br>$5 < \eta \leq 10$<br>$10 < \eta \leq 15$<br>$\eta > 15$ | 1<br>$1-(x-0.8)/0.3$<br>0 | $0 < Ds \leq 0.8$<br>$0.8 < Ds \leq 1.1$<br>$Ds > 1.1$ |
| V2 | 1<br>$1-(x-10)/10$<br>$(x-20)/10$<br>$1-(x-30)/10$<br>0 | $0 < D \leq 10$<br>$10 < D \leq 20$<br>$20 < D \leq 30$<br>$30 < D \leq 40$<br>$D > 40$ | 1<br>$1-(x-5)/5$<br>$(x-10)/5$<br>$1-(x-15)/5$<br>0 | $0 < \eta \leq 5$<br>$5 < \eta \leq 10$<br>$10 < \eta \leq 15$<br>$15 < \eta \leq 20$<br>$\eta > 20$ | 0<br>$(x-0.8)/0.3$<br>$1-(x-1.1)/0.3$<br>0 | $0 < Ds \leq 0.8$<br>$0.8 < Ds \leq 1.1$<br>$1.1 < Ds \leq 1.4$<br>$Ds > 1.4$ |
| V3 | 0<br>$(x-30)/10$<br>$1-(x-40)/10$<br>0 | $0 < D \leq 30$<br>$30 < D \leq 40$<br>$40 < D \leq 50$<br>$D > 50$ | 0<br>$(x-15)/5$<br>$1-(x-20)/5$<br>0 | $0 < \eta \leq 15$<br>$15 < \eta \leq 20$<br>$20 < \eta \leq 25$<br>$\eta > 25$ | 0<br>$(x-1.1)/0.3$<br>$1-(x-1.4)/0.3$<br>0 | $0 < Ds \leq 1.1$<br>$1.1 < Ds \leq 1.4$<br>$1.4 < Ds \leq 1.7$<br>$Ds > 1.7$ |
| V4 | 0<br>$(x-40)/10$<br>1 | $0 < D \leq 40$<br>$40 < D \leq 50$<br>$D > 50$ | 0<br>$(x-20)/5$<br>1 | $0 < \eta \leq 20$<br>$20 < \eta \leq 25$<br>$\eta > 25$ | 0<br>$(x-1.4)/0.3$<br>1 | $0 < Ds \leq 1.4$<br>$1.4 < Ds \leq 1.7$<br>$Ds > 1.7$ |

**Table 12.** Judgment of roughness and micro hardness of each sample.

| Fuzzy Performance Evaluation | Membership Function | Roughness Range | Membership Function | Hardness Range |
|---|---|---|---|---|
| V1 | 1<br>$1-(x-8)/8$<br>0 | $0 < Ra \leq 8$<br>$8 < Ra \leq 16$<br>$Ra > 16$ | 1<br>$(x-840)/60$<br>0 | $Hv \geq 900$<br>$840 \leq Hv < 900$<br>$Hv < 840$ |
| V2 | 0<br>$(x-8)/8$<br>$1-(x-16)/8$<br>0 | $0 < Ra \leq 8$<br>$8 < Ra \leq 16$<br>$16 < Ra \leq 24$<br>$Ra > 24$ | 0<br>$1-(x-840)/60$<br>$(x-780)/60$<br>0 | $Hv \geq 900$<br>$840 \leq Hv < 900$<br>$780 \leq Hv < 840$<br>$Hv < 780$ |
| V3 | 0<br>$(x-16)/8$<br>$1-(x-16)/8$<br>0 | $0 < Ra \leq 16$<br>$16 < Ra \leq 24$<br>$24 < Ra \leq 32$<br>$Ra > 32$ | 0<br>$1-(x-780)/60$<br>$(x-720)/60$<br>0 | $Hv \geq 840$<br>$780 \leq Hv < 840$<br>$720 \leq Hv < 780$<br>$Hv < 720$ |
| V4 | 0<br>$(x-24)/8$<br>1 | $0 < Ra \leq 24$<br>$24 < Ra \leq 32$<br>$Ra > 32$ | 0<br>$1-(x-720)/60$<br>1 | $Hv \geq 780$<br>$720 \leq Hv < 780$<br>$Hv < 720$ |

### 3.3.5. Establishment of Membership Matrix R

A stepwise membership function was used to judge the second-level evaluation index set $Z_j$ of the index layer. After establishing the second-level fuzzy comprehensive judgment matrix, the first-level evaluation index set $B_i$ of the criterion layer was judged. The first-level fuzzy comprehensive judgment matrix was established as follows:

$$R_i = \begin{bmatrix} r_{11} & r_{12} & \cdots & r_{1m} \\ r_{21} & r_{22} & \cdots & r_{2m} \\ \vdots & \vdots & \cdots & \vdots \\ r_{k1} & r_{k2} & \cdots & r_{km} \end{bmatrix}, (i = 1, 2, 3, m = 1, 2, 3), \quad (5)$$

where $r_{km}$ represents the degree of membership between the kth second-level evaluation index and the i-th first-level evaluation index at the m-th quality level.

3.3.6. Fuzzy Comprehensive Evaluation

The comprehensive score vector of the grade i evaluation index was:

$$B_i = B_i \cdot R_i, \ (i = 1, 2, 3), \tag{6}$$

Evaluation index:

$$R = \begin{bmatrix} B_1 & B_2 & B_3 \end{bmatrix}^T, \tag{7}$$

Final composite score vector B:

$$U = W \cdot R, \tag{8}$$

The final quality grade was determined according to the maximum membership principle.

**4. Evaluation Methodology**

*4.1. Computation of the Final Vector*

The quality level of the indicator layer and the final quality level of each sample were determined according to the maximum membership principle. Samples of the same quality level were ranked by comparing the criterion layer membership distribution. The measured second-level evaluation indexes of all samples were placed in Tables 10–12 to obtain the second-level evaluation index of the j-th sample of the membership matrix $R_i(j)$ (i = 1, 2, 3; j = 1, 2, . . ., 25) of the index layer. The matrix $R_i(j)$) in Table 3 and the second-level weight vector of the index layer were introduced into Equation (6) to calculate the composite fuzzy matrices $B_1$, $B_2$, and $B_3$ of the index layer. After combination, the criterion layer composite fuzzy matrix U was calculated by combining the first level weight of the criterion layer. The overall score $G_B$ was obtained by assigning V = (4, 3, 2, 1) to the different levels.

4.1.1. Composite Fuzzy Matrix and Index Results of the Criterion Layer

• Degree of forming quality:

$$
B_1 = 
\begin{bmatrix}
B_1(1) \\
B_1(2) \\
B_1(3) \\
B_1(4) \\
B_1(5) \\
B_1(6) \\
B_1(7) \\
B_1(8) \\
B_1(9) \\
B_1(10) \\
B_1(11) \\
B_1(12) \\
B_1(13) \\
B_1(14) \\
B_1(15) \\
B_1(16) \\
B_1(17) \\
B_1(18) \\
B_1(19) \\
B_1(20) \\
B_1(21) \\
B_1(22) \\
B_1(23) \\
B_1(24) \\
B_1(25)
\end{bmatrix}
=
\begin{bmatrix}
0.8358 & 0.1642 & 0 & 0 \\
0 & 0 & 0.124 & 0.876 \\
0.069 & 0.8122 & 0.1188 & 0 \\
0.4908 & 0.3092 & 0 & 0.2 \\
0.0976 & 0.1948 & 0.5296 & 0.178 \\
0.3408 & 0.3252 & 0.2716 & 0.0624 \\
0.5049 & 0.4951 & 0 & 0 \\
0.2278 & 0.7062 & 0 & 0 \\
0.1989 & 0.8011 & 0 & 0 \\
0 & 0.1692 & 0.1981 & 0.6351 \\
0.384 & 0.216 & 0.0816 & 0.3184 \\
0.6679 & 0.3071 & 0.025 & 0 \\
0.0172 & 0.7828 & 0.0659 & 0.1341 \\
0 & 0.118 & 0.2777 & 0.6043 \\
0.1302 & 0.2414 & 0.4284 & 0.2 \\
0.3444 & 0.2556 & 0 & 0.4 \\
0.4248 & 0.4583 & 0.1169 & 0 \\
0 & 0.8217 & 0.1783 & 0 \\
0.0486 & 0.5996 & 0.3518 & 0 \\
0.7672 & 0.0937 & 0.1391 & 0 \\
0.57 & 0.03 & 0 & 0.4 \\
0 & 0.162 & 0.4928 & 0.3452 \\
0.1066 & 0.0934 & 0.3 & 0.5 \\
0.2772 & 0.7028 & 0.02 & 0 \\
0.5978 & 0.287 & 0.1152 & 0
\end{bmatrix}
$$

A sufficient thickness provides cutting height for subsequent machining, while a sufficiently low dilution rate minimizes atomic diffusion and thermal effects on the substrate, and a slower cladding speed produces a thicker cladding layer. The evaluation results in terms of forming quality are shown in Figure 11. The excellent-quality samples included

samples 1, 20, 12, 25, 21, 7, 4, 11, and 6. Good-quality samples included 18, 3, 9, 13, 8, 24, 19, and 17. Average-quality samples were 5, 22, and 15. Poor-quality samples were 2, 10, 14, 23, and 16. In contrast to traditional laser cladding, the scanning speed had a much greater effect on the coating thickness than the powder feed rate during high-speed laser cladding. The overlap rate had the greatest influence on the dilution rate, which is consistent with Qiao's findings [41].

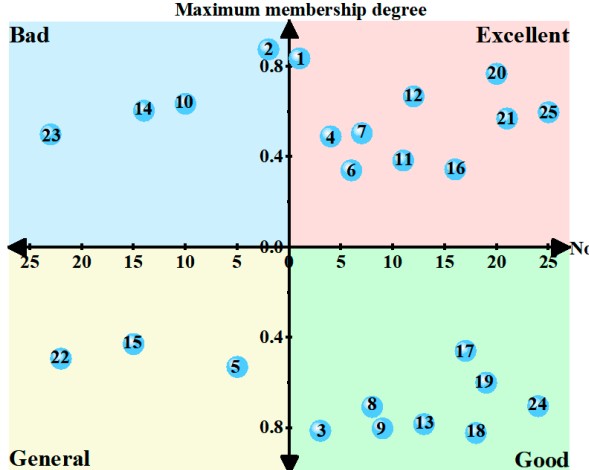

**Figure 11.** Evaluation result of coating forming quality ($B_1$).

- Microstructure:

$$
B_2 = \begin{bmatrix} B_2(1) \\ B_2(2) \\ B_2(3) \\ B_2(4) \\ B_2(5) \\ B_2(6) \\ B_2(7) \\ B_2(8) \\ B_2(9) \\ B_2(10) \\ B_2(11) \\ B_2(12) \\ B_2(13) \\ B_2(14) \\ B_2(15) \\ B_2(16) \\ B_2(17) \\ B_2(18) \\ B_2(19) \\ B_2(20) \\ B_2(21) \\ B_2(22) \\ B_2(23) \\ B_2(24) \\ B_2(25) \end{bmatrix} = \begin{bmatrix} 0 & 0 & 0 & 1 \\ 0 & 0 & 0.7962 & 0.2038 \\ 1 & 0 & 0 & 0 \\ 1 & 0 & 0 & 0 \\ 0.755 & 0.245 & 0 & 0 \\ 0 & 0 & 0.6712 & 0.3288 \\ 0 & 0 & 0.9 & 0.1 \\ 1 & 0 & 0 & 0 \\ 1 & 0 & 0 & 0 \\ 0 & 0 & 0.9937 & 0.0063 \\ 0 & 0 & 0.7812 & 0.2188 \\ 0 & 0.7625 & 0.2375 & 0 \\ 0.6562 & 0.3438 & 0 & 0 \\ 0.2175 & 0.7825 & 0 & 0 \\ 0 & 0.225 & 0.775 & 0 \\ 0 & 0.6375 & 0.3625 & 0 \\ 0 & 0.3887 & 0.6113 & 0 \\ 0 & 0.3275 & 0.6725 & 0 \\ 0 & 0.6812 & 0.3188 & 0 \\ 0 & 0.6475 & 0.3525 & 0 \\ 0 & 0 & 0.6877 & 0.3213 \\ 0 & 0 & 0.6877 & 0.3213 \\ 0 & 0.7937 & 0.2063 & 0 \\ 1 & 0 & 0 & 0 \\ 1 & 0 & 0 & 0 \end{bmatrix}
$$

The micro-structure evaluation results obtained by the composite fuzzy matrix $B_2$ are shown in Figure 12. An excellent micro-structure was obtained for 3, 4, 8, 9, 24, 25, 5, and 13. Good samples were 23, 14, 12, 19, 20, and 16. Average samples were 10, 7, 2, 11, 18, 6, 21, 22, 17, and 15. The worst sample was number 1. The scanning speed greatly densified

the micro-structure, and the high overlap rate homogenized the near-surface dendrites. Zhang's results back this up [42].

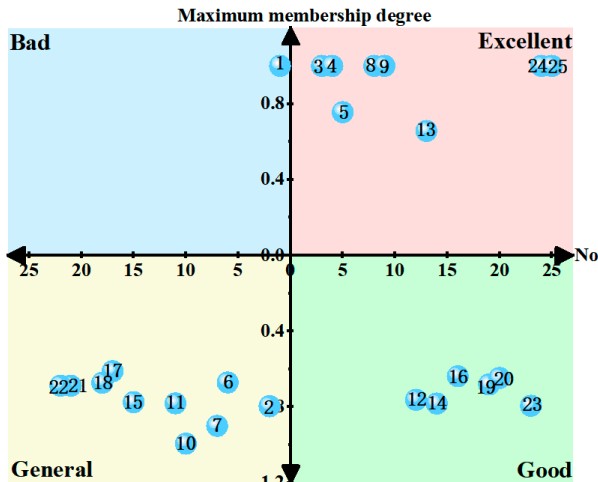

**Figure 12.** Evaluation result of coating micro-structure (B$_2$).

- Surface properties:

$$
B_3 = \begin{bmatrix} B_3(1) \\ B_3(2) \\ B_3(3) \\ B_3(4) \\ B_3(5) \\ B_3(6) \\ B_3(7) \\ B_3(8) \\ B_3(9) \\ B_3(10) \\ B_3(11) \\ B_3(12) \\ B_3(13) \\ B_3(14) \\ B_3(15) \\ B_3(16) \\ B_3(17) \\ B_3(18) \\ B_3(19) \\ B_3(20) \\ B_3(21) \\ B_3(22) \\ B_3(23) \\ B_3(24) \\ B_3(25) \end{bmatrix} = \begin{bmatrix} 0 & 0 & 0.4984 & 0.5016 \\ 0 & 0 & 0.8138 & 0.1862 \\ 1 & 0 & 0 & 0 \\ 1 & 0 & 0 & 0 \\ 0.9184 & 0.0816 & 0 & 0 \\ 0 & 0 & 0.2235 & 0.7765 \\ 0 & 0.6551 & 0.3116 & 0.0333 \\ 0.5859 & 0.4141 & 0 & 0 \\ 0.333 & 0.4748 & 0.1922 & 0 \\ 0 & 0.2557 & 0.7422 & 0.0021 \\ 0.2633 & 0.4037 & 0.2601 & 0.0729 \\ 0 & 0.6978 & 0.3022 & 0 \\ 0.4176 & 0.5824 & 0 & 0 \\ 0.0724 & 0.4581 & 0.4694 & 0 \\ 0.4069 & 0.3351 & 0.2581 & 0 \\ 0 & 0.2123 & 0.1207 & 0.667 \\ 0 & 0.3766 & 0.6234 & 0 \\ 0 & 0.1091 & 0.6306 & 0.2603 \\ 0 & 0.5743 & 0.4257 & 0 \\ 0 & 0.6841 & 0.3159 & 0 \\ 0 & 0 & 0.229 & 0.774 \\ 0 & 0 & 0.3374 & 0.6656 \\ 0 & 0.2643 & 0.63 & 0.1057 \\ 0.333 & 0 & 0.5732 & 0.0938 \\ 0.4386 & 0.5615 & 0 & 0 \end{bmatrix}
$$

Surface roughness and hardness are the main factors affecting the actual working ability of a workpiece. The surface roughness of the high-speed laser cladding coating was less than 1/10 of traditional cladding. The evaluation results of the coating surface performance of B3 are shown in Figure 13. The excellent grades included 3, 4, 5, and 8. The good samples were 12, 20, 7, 13, 19, 25, 9, and 11. The average samples were 2, 10, 18, 17, 23, 14, and 24. The samples with poor quality were 6, 21, 16, 22, and 1. Due to the high overlap rate of high-speed laser cladding, the correlation between the coating surface roughness and overlap rate was lower than that of traditional cladding. Moreover, due to

its two-stage melting characteristics, the laser energy was blocked by the powder, which reduced the thermal effects on the substrate [43]. There was a stronger correlation between the surface roughness of the coating and the feed rate, and increases in the coating hardness were consistent with grain refinement near the surface.

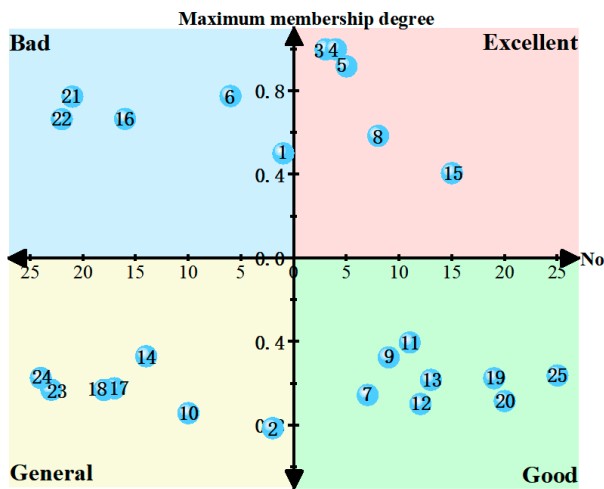

**Figure 13.** Evaluation result of coating surface properties (B₃).

### 4.1.2. Composite Fuzzy Matrix and Index Results of the Criterion Layer

$$
U = \begin{bmatrix} B(1) \\ B(2) \\ B(3) \\ B(4) \\ B(5) \\ B(6) \\ B(7) \\ B(8) \\ B(9) \\ B(10) \\ B(11) \\ B(12) \\ B(13) \\ B(14) \\ B(15) \\ B(16) \\ B(17) \\ B(18) \\ B(19) \\ B(20) \\ B(21) \\ B(22) \\ B(23) \\ B(24) \\ B(25) \end{bmatrix} = \begin{bmatrix} 0.1141 & 0.0224 & 0.3115 & 0.552 \\ 0 & 0 & 0.5255 & 0.4745 \\ 0.6901 & 0.2937 & 0.0162 & 0 \\ 0.9305 & 0.0422 & 0 & 0.0273 \\ 0.8258 & 0.0776 & 0.0723 & 0.0243 \\ 0.0465 & 0.0444 & 0.1768 & 0.7323 \\ 0.0689 & 0.7155 & 0.1947 & 0.0208 \\ 0.6358 & 0.3552 & 0 & 0 \\ 0.4022 & 0.4776 & 0.1201 & 0 \\ 0 & 0.1908 & 0.7215 & 0.088 \\ 0.217 & 0.4805 & 0.2135 & 0.089 \\ 0.274 & 0.5337 & 0.1923 & 0 \\ 0.3826 & 0.5901 & 0.009 & 0.0183 \\ 0.0453 & 0.3024 & 0.4823 & 0.1699 \\ 0.2721 & 0.3855 & 0.3152 & 0.0273 \\ 0.047 & 0.3584 & 0.1231 & 0.4715 \\ 0.0659 & 0.5285 & 0.4056 & 0 \\ 0 & 0.1803 & 0.4185 & 0.4012 \\ 0.0066 & 0.5601 & 0.4333 & 0 \\ 0.1763 & 0.6073 & 0.2165 & 0 \\ 0.0778 & 0.0756 & 0.3101 & 0.5383 \\ 0 & 0.0221 & 0.2781 & 0.7016 \\ 0.0146 & 0.2733 & 0.5778 & 0.1343 \\ 0.3016 & 0.2788 & 0.361 & 0.0587 \\ 0.5942 & 0.3901 & 0.0157 & 0 \end{bmatrix}, G_B = \begin{bmatrix} G_B(1) \\ G_B(2) \\ G_B(3) \\ G_B(4) \\ G_B(5) \\ G_B(6) \\ G_B(7) \\ G_B(8) \\ G_B(9) \\ G_B(10) \\ G_B(11) \\ G_B(12) \\ G_B(13) \\ G_B(14) \\ G_B(15) \\ G_B(16) \\ G_B(17) \\ G_B(18) \\ G_B(19) \\ G_B(20) \\ G_B(21) \\ G_B(22) \\ G_B(23) \\ G_B(24) \\ G_B(25) \end{bmatrix} = \begin{bmatrix} 1.7 \\ 1.53 \\ 3.67 \\ 3.88 \\ 3.7 \\ 1.41 \\ 2.83 \\ 3.61 \\ 3.28 \\ 2.1 \\ 2.83 \\ 3.08 \\ 3.34 \\ 2.22 \\ 2.9 \\ 1.98 \\ 2.66 \\ 1.78 \\ 2.57 \\ 2.96 \\ 1.7 \\ 1.32 \\ 2.17 \\ 2.82 \\ 3.58 \end{bmatrix}
$$

The coating quality evaluation results based on the maximum membership principle are shown in Figure 14. The coatings with excellent comprehensive quality were 3, 4, 5, 8, 24, and 25. The good samples were 7, 9, 11, 12, 13, 15, 17, 19, and 20. The average samples were 2, 10, 14, 18, and 23. The samples with poor quality were 1, 6, 16, 21, and 22.

Combining the maximum membership map with the final score shows that sample 4 had the best quality, and sample 6 had the worst quality.

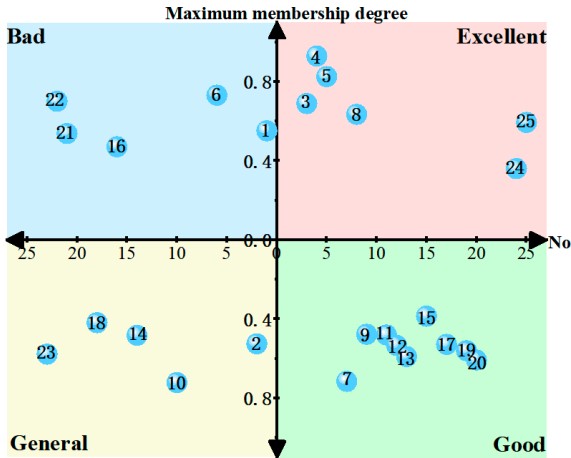

**Figure 14.** Evaluation result of coating quality (U).

*4.2. Machining Parameter Optimization*

Fuzzy comprehensive evaluation analysis was combined with the assigned grades to obtain the comprehensive score of each sample. From this, the optimal process parameters of high-speed laser cladding of a Fe-Cr-Ni based alloy coating were determined to be a laser power of 660 W, a scanning speed of 14,400 mm/min, a lap rate of 65%, and a powder feeding rate of 4 r/min.

**5. Discussion**

In this paper, high-speed laser cladding experiments were carried out using ASTM 1045 steel and an iron-based self-melting alloy powder. The AHP-FCE method was used to evaluate the quality of high-speed laser cladding multi-path overlap surface, and the relationship between the high-speed laser cladding process parameters and coating quality was established to guide the actual operation. The validity of the model was proved by mutual verification between the evaluation results and experimental data. However, the research in this paper has some limitations: 1. The single-layer thickness of the high-speed laser cladding coating was only tens to hundreds of microns. Various applications require the preparation of multiple layers, which increases the coating uncontrollability of the macro- and microstructure and performance. The proposed method will be used to explore the preparation of multilayer coatings in future work. 2. In this paper, thin coatings (with only tens of microns in a single layer) of some samples displayed large errors when testing their wear and corrosion properties, showing that this method is insufficiently accurate to evaluate their surface properties. In future work, a more in-depth performance evaluation study will be carried out on multi-layer coating surfaces. 3. The influence of different processing techniques on the coating quality of only one type of material was analyzed, and the model must be extended to evaluate the coating performance of other materials.

**6. Conclusions**

In this paper, 25 groups of parameters were selected to experimentally analyze the influence of processing parameters on the forming quality, microstructure, and surface properties of high-speed laser cladding coatings. We established an AHP-FCE comprehensive evaluation model of high-speed laser cladding coating quality, and the membership function calculation and fuzzy evaluation ranking of the indexes were carried out according to the maximum membership principle. The coating surface quality was quantitatively evaluated, and the artificial subjectivity and incoherent data correspondence were avoided. This solved the problem in which current quality evaluations of high-speed laser cladding

coatings are not systematic or scientific and provides decision-making ideas for actual repair and processing. Based on the results, the following conclusions were drawn:

(1) The analytic hierarchy process (AHP) enables an objective determination of the contribution made by each factor in the criterion layer and index layer to coating quality. In the criterion layer, the evaluation weights for formability, microstructure, and surface properties are 13.65%, 23.85%, and 62.5%, respectively. Dilution rate is identified as the primary factor affecting formability (60%), while microhardness is identified as the main factor influencing surface properties (66.7%). By utilizing fuzzy comprehensive evaluation (FCE), a multi-objective quality evaluation process can be transformed into a single objective problem with the weight of criterion layer serving as a weighting to establish graded evaluation levels based on detection results from each index judged by membership function.

(2) Analyze the influence of process parameters on each index: high-speed laser cladding scanning speed has a significantly greater effect on coating thickness than powder feeding speed, while overlap rate has the most significant impact on dilution rate. Scanning speed greatly enhances microstructure densification, whereas high binding rates result in homogenization of near-surface dendrites. There is a significant correlation between surface roughness of coatings and powder feeding rates; an increase in coating hardness corresponds with observed thinning trends in near-surface grain size.

(3) Through fuzzy comprehensive evaluation, we have established a theoretical analysis model for evaluating high-speed laser cladding surface quality that identifies optimum preparation processes based on contributions made by formability, microstructure, and surface properties: laser power at 660 W; scanning speed at 14,400 mm/min; overlap rate at 65%; powder feeding speed at 4 r/min. Finally, the comprehensive quantification of coating quality is realized, and a new method for quantitative evaluation and visualization of coating quality is provided.

**Author Contributions:** Conceptualization, Y.X.; methodology, W.C.; software, Y.X.; validation, Y.X.; formal analysis, Y.X.; investigation, Y.Z.; resources, Y.S.; data curation, Y.Z.; writing—original draft preparation, Y.X.; writing—review and editing, Y.S.; visualization, Y.Z.; supervision, Y.Z.; project administration, Y.S.; funding acquisition, Y.S. All authors have read and agreed to the published version of the manuscript.

**Funding:** This research was funded by Xinjiang Uygur Autonomous Region Key Research and Development Task Special Project, grant number 2022B01036-1 and Xinjiang Uygur Autonomous Region Central Government Guide Local Science and Technology Development Fund Project, grant number ZYYD2023B03.

**Institutional Review Board Statement:** Not applicable.

**Informed Consent Statement:** Not applicable.

**Data Availability Statement:** There is no additional data reported in this study to support the results.

**Conflicts of Interest:** The authors declare no conflict of interest.

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
