# Peer review of "Engineering Process Optimization and Quality Stability Control of High-Speed Laser Cladding Coatings Based on AHP-FCE"

_coatings, doi:10.3390/coatings13101806_

Round 1
Reviewer 1 Report
The manuscript is original and well-written. I note only two little comments
1. Increase the contrast of the fig.3 (d) (the insert)
2. Do not put the doi reference (in the references list) in square brackets.
Reviewer 2 Report
The presented manuscript “Engineering process optimization and quality stability control of high-speed laser cladding coatings based on AHP-FCE” contains interesting results. However, I think that this manuscript requires some major improvements in following areas:
· Abstract needs to be revised. I didn’t observed any quantitative finding in abstract. It must be added. It should address a) need of this work, b) novelty, c) methodology suggested, d) results obtained (numerical).
· Introduction section is again very poorly written. Revise it and add the relevant information with proper flow. Avoid bulk citations
· I didn’t find relevant literature work to justify the research gap. Relevant literature must be added to justify the research gap and need of the present study.
· In the last paragraph of introduction, write the novelty of your work and selected conditions for experimentations.
· Specifications of the used equipment should be presented.
· Add the figure of the experimental setup.
· Table 2: What is the basis of the selection of mentioned set of parameters?
· Specify the upper and lower bounds of the used experimental setup
· Results and discussion section is written well. However, compare the obtained results with similar past studies and justify the findings with proper technical reasons in results and discussion section.
· Conclusion section need to manage properly in key points.
Reviewer 3 Report
In this manuscript, the authors studied the engineering process optimization and quality stability control of high-speed laser cladding coatings based on AHP-FCE. The manuscript should be accepted after addressing the following issues;
1) In materials and method, there is no information about the materials source and purification etc. The authors should provide this information.
2) In the material and method section, the authors mentioned that “An ASTM 1045 steel plate with dimensions of 70 mm×150 mm×8 mm was used as the 61 substrates.” Did the authors clean the plate before use as a substrate? Is there any effect of other substrates?
3) What’s about the defects in the thin films?
4) Did the authors check the elemental composition, after the deposition, sometime the other elements were added as impurities.
5) What specific improvements should the authors consider regarding the
methodology? What further controls should be considered?
6) What you did in this article should be written in the abstract. The reader is confused about ``what you actually did or what is your part in this part. They should improve the abstract and conclusions.
7) The authors should mention in the manuscript how this work is different from the previous work.
8) In Figure 3, the related information was hard to read; the authors should improve it.
9) There are many spelling and formatting typos in this paper; the authors should check and revise them thoroughly
10) What specific improvements should the authors consider regarding the methodology? What further controls should be considered?
11) Are the conclusions consistent with the evidence and arguments presented, and do they address the main question posed?
Moderate editing of English language required
Reviewer 4 Report
It is recommended to accept after minor revision this manuscript (coatings-2636793). Also, there are some (4) notes/corrections.
A. List of (2) major notes/corrections:
1) Line_002. Illuminate a term (stability). It is recommended to illuminate the term 'stability control' (and keyword, 'Coating quality stability
control') found in the title, by adding illustrative (con)text in the paragraph of the discussion (4. Discussion) paragraph. Otherwise, modify the title, in: "Engineering process optimization and quality stability control".
2) Line_134. Explain the parameter 'Ra' (and 'Or') found in table 3, in: "Table 3. Surface microscopic morphology and Roughness of high-speed laser cladding coating.".
B. List of (2) minor notes/corrections:
1) Line_069. Delete a sub-figure (#13). It is recommended to delete the (second) sub-figure, 'macroscopic morphology #13', of the figure 1, in: "Figure 1. Macroscopic morphology of high-speed laser cladding coating.".
2) Line_133. Re-arrange syntax. Re-arrange the syntax of 'Dendrite size and microhardness' to be 'Microhardness and dendrite size', in Y1-Y2 axis labels, at description of the figure 5, in: "Figure 5. Dendrite size and microhardness of high-speed laser cladding coating near surface position".
Reviewer 5 Report
In my opinion, a very interesting article. The research results can be used industrially, which is a great advantage of the work. The optimization carried out enables the control of the high-speed laser cladding process. Work defects that can be easily repaired include:
1. No photo of the research position.
2. Lack of characteristics of the laser device. This is its individual feature and the results obtained are valid only for this device. However, the methodology is universal and can be applied to other laser devices.
Round 2
Reviewer 3 Report
Accepted in the present form
Minor editing of English language required